# Processing of Low-Density HGM-Filled Epoxy–Syntactic Foam Composites with High Specific Properties for Marine Applications

**DOI:** 10.3390/ma16041732

**Published:** 2023-02-20

**Authors:** Olusegun Adigun Afolabi, Turup Pandurangan Mohan, Krishnan Kanny

**Affiliations:** Composite Research Group, Department of Mechanical Engineering, Durban University of Technology, Durban 4000, South Africa

**Keywords:** epoxy resin, hollow glass microspheres, syntactic foam composites, density, packing efficiency, specific tensile strength, wall thickness, aspect ratio

## Abstract

A solution casting approach is used to create hollow glass microsphere (HGM)-filled epoxy–syntactic foam composites (e–SFCs) by varying the concentrations of HGM in epoxy according to different particle sizes. Density analysis is used to investigate the impact of concentration and particle size regularity on the microstructure of e-SFCs. It was observed that e–SFCs filled with an HGM of uniform particle sizes exhibit a reduction in density with increasing HGM concentration, whereas e-SFCs filled with heterogeneous sizes of HGM exhibit closeness in density values regardless of HGM concentration. The variation in e–SFC density can be related to HGM packing efficiency within e–SFCs in terms of concentration and particle size regularity. The particle size with lowest true density of 0.5529 g/cm^3^, experimental density of 0.949 g/cm^3^ and tensile strength of 55.74 MPa resulted in e-SFCs with highest specific properties of 100.81 (MPa·g/cm^3^), with a 35.1% increase from the lowest value of 74.64 (MPa·g/cm^3^) at a true density of 0.7286 g/cm^3^, experimental density of 0.928 g/cm^3^ and tensile strength of 54.38 MPa. The e–SFCs’ theoretical density values were obtained. The variance in theoretical and experimental density values provides a thorough grasp of packing efficiency and inter-particle features.

## 1. Introduction

The relative density (RD) of composite materials is used in composite materials to identify them, track physical changes, and ensure acceptable product quality [1]. Syntactic foam (SF) as a composite (SFC) material is a lightweight material made by incorporating hollow glass microballoons (HGMs) into a suitable polymer, aluminum, or metal matrix. The incorporation of HGMs into the matrix has a considerable impact on the weight of polymer materials, making them appropriate for a variety of industrial and structural applications in aircraft, maritime, and building partitioning. SFC materials are lighter in weight, have a greater specific strength and have a lower coefficient of thermal expansion than the basic matrix [2,3,4]. The SFC properties are determined by particle size and concentration of HGM particles. As a result, it is crucial to figure out and analyze the impact of particle size and concentration on the density of e–SFC material. 

The density of a material is defined as the mass per unit volume ratio of that material. Because density is often the single parameter that is most clearly related to the physical and mechanical properties of polymers and polymer composites, it has been used to identify polymers, track physical changes in polymers and polymer composites, and ensure product quality for suppliers and processors [1]. The physical concept of density is used to understand the impacts of composite material composition and mechanical qualities. 

The density of composite materials forms most of the physical basis of the method of characterizing the effects of mechanical properties and determining their positioning in composite materials [5]. SFCs have been studied to exhibit the same density before and after curing, making them suitable in aerospace applications [6,7]. SFCs exhibit differences in density because of the difference in the internal radius and not the outer radius; this was studied by Woldesenbet et al. and Afolabi et al. [8,9], which resulted in a considerable increase in the peak strength of a SFC for higher strain rates and increased density. This paper aims to improve our understanding of the effects of concentration and particle size on density. Musha et al. [10] previously reported that size and density differences are some of the major factors that influence the mixing properties of composite materials. Additionally, Ozkutlu et al. [11], gave a report on the influence of HGM density on the mechanical and thermal properties of poly (methyl methacrylate) (PMMA) syntactic foam composites. It was observed that higher density HGMs remained intact in the PPMA matrix and increasing HGM density resulted in a reduction in mechanical properties. SFC as a form of polymeric composite materials (PCMs) typically possess numerous properties, such as low density, good thermal insulation, low coefficient of thermal expansion, high strength-to-weight ratio, good acoustic properties, low moisture absorption and good resistance to hydrostatic pressure [12,13,14,15,16]. Most of these special properties are caused by various factors, such as filler sizes, type of binder used, volume fraction considered, void fraction, filler/matrix interface and condition of manufacturing, which yields unique applications [8]. These properties have enhanced their applications as fireproof foam [12], soundproofing materials to prevent noise pollution in structural and marine applications [17], as well as in heat resistance, thermal resistance and thermal stability of composites [18], etc. 

However, there has not been any report on the influence of low-density HGMs at a higher particle size on the specific tensile properties of e–SFCs. Therefore, in this paper, e–SFCs (epoxy-SFCs) were made with epoxy resin and HGMs (at various concentrations and particle sizes) for tensile and specific tensile strength evaluation, and their density was determined using ASTM D792-08, Standard Test Methods for Density and Specific Gravity (Relative Density) of Plastics by Displacement [19]. The effect of low density with high specific properties and the importance of wall thickness and aspect ratio on e–SFCs for marine applications is also considered. 

## 2. Materials and Methods

### 2.1. Materials

In the fabrication of e–SFCs, a low-viscosity epoxy resin (LR20) and hardener (LR281), as well as HGM (grade T60), were utilized without any further treatment. They were bought from AMT Composites in South Africa and Anhui Elite Industrial Cop, Limited, Hong Kong Elite Industrial Group Limited in China, respectively.

The HGM had a density of 0.6 g/cm^3^ with particle sizes of 10–60 µm. The physiochemical characteristics and viscosity of the constituent components were discussed in our previous work [9]. The HGM was varied into four particle sizes using different sieve ranges and characterized by the particle distribution analysis (PSA) detailed process was reported earlier by the author [20]. Furthermore, particle variation analysis was conducted using a gas pycnometer (Ultrapyc 5000 with 19 psi pressure by Anton Paar, Graz, Austria) to determine the densities of the varied sizes. Figure 1 shows the schematic representation of the HGM before and after particle variation analysis. 

### 2.2. Methods

#### Fabrication of e–SFCs

Before adding the HGM, LR20 was heated to 60 °C for 60 min and then cooled to room temperature. Using a high-speed stirrer for 10 min, homogeneous dispersion of the HGM into LR20 was optimized. The viscosity changes experienced by the addition of HGM is presented in our previous work [9]. Later, LR281 was added to a homogeneous suspension of LR20-HGM and mixed for 5 min. The uniform LR20-HGM-LR281 mixture was put into the silicone mold and allowed to cure for 24 h at room temperature (27 °C). The e–SFCs were then post-cured at 80 °C for 240 min for better specimen solidification before testing. Table 1 shows more information about the e–SFCs’ formulation.

### 2.3. Denaisty Measurement of e–SFCs

Experimental density for e–SFCs was determined using ASTM D792-08, Standard Test Methods for Density and Specific Gravity (Relative Density) of Plastics by Displacement. The theoretical density of the e–SFCs was determined using Equation (1) [21].
(1)$$\rho_{p}=\frac{\rho_{c}-\rho_{f}\times {\;V}_{f}}{1-V_{f}}$$
where ${\;\rho }_{p}$, $\rho_{c}$, $\rho_{f}$, and $V_{f}$ are the density of the polymer matrix, composite, filler, and volume fraction of filler, respectively.

Table 2 show the comparison of different characteristics properties of e–SFCs at 5 vol% for the varied particle sizes. The 5 vol% was considered the lowest volume fraction because we are focusing on the lowest true density. Additionally, it was previously reported that good adhesion occurs in composite materials at a lower volume fraction [9,20]. The average wall thickness “t” is a measure of strength distribution and the aspect ratio “a”, which influences the strength of HGM particles, and was calculated using Equations (2) and (3), respectively [22].
(2)t=d2 1−1−ρuρs13
(3)$$a=\frac{d}{t}$$
where d (as earlier reported by the authors [20]), $\rho_{u}$, and $\rho_{s}$ are the average diameters of the particle distribution of T60-HGM, the average true density of particle distribution of T60-HGM, and the density of the wall material (2.56 g/cm^3^), respectively.

### 2.4. Mechanical Characterization

The tensile analyses of e–SFCs were carried out according to ASTM D 3039 test standard specifications. The test was carried out on MTS 793 servo-hydraulic machine with a load cell of 30 KN. Detailed test parameters were discussed earlier by the author [9].

### 2.5. Scanning Electron Microscopy

The fracture surface of the e–SFCs were examined by scanning electron microscopy (SEM), using a Zeiss EVO 1 HD 15 Oxford instrument X-max. The tensile fracture specimens were used for SEM analysis, and they were gold-coated prior to the analysis using a Quorum Q 150R ES machine [20].

## 3. Results and Discussion

Figure 2a–t show the schematic microstructure of the e–SFCs filled with uniform HGM particle sizes, while Figure 2u–y show the microstructure of the e–SFCs filled with heterogeneous HGM particle sizes. It could be noticed that the uniform HGM particle sizes consisted of closer particle ranges (i.e., 20–24 µm, to 50–60 µm). This was responsible for the few different circles in these sizes. 

We graphically represent the density values of the e–SFCs filled with uniform particle size and varying concentration versus neat epoxy (NE) in Figure 3, and uniform concentration and varying particle sizes in Figure 4, to gain a better understanding of particle sizes versus the concentration of density of e–SFCs. Figure 3a–d show a drop in density values for e–SFCs with a uniformly sized HGM (irrespective of HGM concentration) when compared to NE, which may be attributed to a decrease in the vol% of epoxy, which corresponds to the substitution with an equivalent vol% of HGM.

The decrease in epoxy matrix vol% and replacement with comparable vol% of HGM creates a porous structure, lowering the density of e–SFCs filled with a uniformly sized HGM [23]. The density values of e–SFCs filled with heterogeneous sizes of HGM are graphically shown in Figure 3e. There is little variation in the density values of e-SFCs filled with heterogeneous sizes of HGM. This may be ascribed to the well-disseminated HGM inside the epoxy matrix, where the gaps between the large-sized HGMs are occupied by tiny-sized HGMs, preventing the spaces from being left empty. This can also be attributed to different particle size properties, which invariably affect mixing behavior of the particles [10]. Furthermore, an increase in either particle size or density can result in the rate of segregation of composite materials. Segregation is caused by the generation of vertical driving force on the particles due to their differences in particle size and density [10,24].

From Figure 4a through Figure 4e, an unusual outcome emerges. Compared to NE and e–SFCs filled with heterogeneous sizes of HGMs, there was a drop in density values for the e–SFCs filled equally with HGMs of particular particle sizes [23]. Second, when the concentration of HGM increases, the density value continues to drop, which can be attributed to the decrease in epoxy matrix vol% and the replacement with a comparable vol% of HGMs, which creates a porous structure, lowering the density of e–SFCs filled with a particular sized HGM. Although the drop in density is exactly proportional to the concentration, there was a small rise in density with increasing particle size within a given concentration. This was due to the development of a high-density interphase zone within the e–SFCs, which grows in size as the particle size increases. 

The graphical depiction of theoretical density versus experimental density is shown in Figure 5a–e. When e–SFCs filled with uniform sizes of HGMs are compared to theoretical density, the experimental density drops; however, when e–SFCs filled with heterogeneous sizes of HGMs are compared to theoretical density, the experimental density rises. The discrepancy in real contact surface and theoretical contact surface predicted by the equation can be ascribed to the modest drop in experimental density for e–SFCs filled with uniform sizes of HGM. The highest packing efficiency, i.e., the maximum real contact surface, which was not anticipated by the theoretical equation, can be related to the rise in experimental density for e–SFCs packed with heterogeneous sizes of HGM. This can be said to be responsible for its reduced porosity (void) in the micrograph images. A similar case was reported by Ding et al. [25], where an HGM was characterized into different particles to reduce its density and increase its mechanical properties. 

Figure 6a–e shows the tensile stress–strain of the e–SFCs compared to the NE. The tensile properties of the e–SFCs increased with the inclusion of the HGM compared to the NE. The curves show a similar stress–strain relationship consisting of linear elastic regions, followed by the specimen’s brittle fracture. It also indicated that the tensile properties of e –SFCs increased upon the inclusion of the HGM compared to the NE. With the addition of the HGM, all the e–SFCs of the homogeneous HGM (AA5—O5, to AA25—OO25) withstood more load before failure than the NE because the presence of the HGM and its synergistic effect as filler reduced the void content, was lightweight, and increased the tensile strength of the SFC as discussed in a previously published article [9]. This was because the presence of the HGM in the matrix increased the adhesion capacity of the SFCs. The strain shows that the elongation of the e–SFCs and the neat epoxy could stretch before failure. The strain of the NE for all the compositions had the highest elongation. The tensile property values were normalized with their experimental densities, which gives us the specific tensile properties as reported in Figure 7a–e. 

Figure 7 shows a graphical representation of the tensile and specific tensile strength of the e–SFCs compared with the neat epoxy at varied particle sizes. Five-volume fractions (5–25%) were considered for comparison with a focus on the 5 vol% as the lowest fraction because stronger adhesion of composites occurred at a smaller volume fraction [26]. The size and shape of the HGM constituted part of the factors that affect the syntactic foam tensile properties. From Figure 7a, it can be seen that at 5 vol%, DD with the lowest true density of 0.5529 g/cm^3^ had the highest specific tensile strength of 100.81 MPa·g/cm^3^, with an increase of 35% higher than BB. However, all the compositions show better tensile performance than the neat epoxy. This is an indication that the addition of the HGM improved the tensile and specific tensile properties of e–SFCs at all levels of particle size variations.

However, the higher value of specific strength obtained at DD indicated that lowering the density can improve the specific tensile strength of e–SFCs. The fractionation of the sizes was an experiment conducted to understand how a reduction in the density of the SFC can influence the improvement in specific tensile strength of the composite. This result is supported by the flexural properties of the SFC reported in our previous work [20]. This claim is also supported by the wall thickness “t” and the aspect ratio “a” of 5.990 µm and 9.188%, respectively, as reported in Table 2. It was also observed that the density (experimental and theoretical in Figure 3 and Figure 5) of the e–SFCs increased at 5 vol% loading, rather than another concentration. This indicated that a reduction in the density of material by particle size will possibly lead to higher specific property of the material, leading to reduced weight, which can be applicable in marine and submarine purposes. 

Figure 7b–e show that as the volume fraction of HGM increases (i.e., from 10% upward), the highest specific tensile strength is at OO with heterogeneous HGM particle sizes, while the varied particle sizes showed inconsistency in their specific tensile strength values. This implies that increasing the volume fraction of HGMs does not correlate to a reduction in the density of the material to increase specific tensile strength. This may be because at higher volume fractions, agglomeration tends to occur during mixing, thereby affecting the mechanical strength of the material [11,22].

Figure 8 shows the SEM micrograph images of the fractured surfaces of the e-SFC tensile specimens compared to the NE. The NE shows a plain surface with matrix porosity, indicating the absence of an HGM. This resulted in their low tensile and specific tensile strength. A fractured HGM and a deboned HGM can be seen on the surfaces of OO–5, AA–5, CC–5, and DD–5, while BB–5 shows more porosity and a rough surface, resulting in reduced specific tensile strength compared to other e–SFCs because of possible poor interaction at that particle size of the HGM. A similar observation was reported by Ozkutly et al. [11], where poor incompatibility between the HGM and the PPMA matrix resulted in lower strength of syntactic foam. The DD–5 reflects good interfacial bonding and adhesion of the HGM and epoxy matrix with reduced porosity and surface roughness, leading to improved specific tensile strength of the e–SFCs. 

## 4. Conclusions

The most significant fundamental qualities of matter are mass and packing, which determine the properties of any material. The aforementioned findings show that the particle size of the HGM affects density in e–FSCs. The density of e–SFCs filled with uniform sizes of the HGM reduced as the concentration of the HGM increased, which can be attributed to interphase organization, in which uniform-sized well-dispersed HGMs occupy the gaps inside the epoxy matrix. The tensile stress–strain of the e-SFCs improved significantly compared to the NE. The reduction in the true density of e–SFCs at DD compared to other sizes showed an increased in experimental density, tensile strength and specific tensile properties of 0.949 g/cm^3^, 55.74 MPa and 100.81 MPa·g/cm^3^, respectively, at 5 vol% with a 2.3%, 2.5% and 35.1% improvement compared to BB and an experimental density of 0.928 g/cm^3^, tensile strength of 54.38 MPa and specific tensile strength of 74.64 MPa·g/cm^3^. This shows that the larger the particle size, the lower the true density. Additionally, the wall thickness “t” and the aspect ratio “a” of DD (5.990 µm and 9.188, respectively) increased with the reduced density compared to the other particle sizes. Meanwhile, with mixed sizes of HGM, the density was nearly constant regardless of HGM concentration, which can be attributed to the well-dispersed HGM inside the epoxy matrix, where the gaps between the large-sized HGM are occupied by tiny-sized HGM, preventing the spaces from being unoccupied. 

## Figures and Tables

**Figure 1 materials-16-01732-f001:**
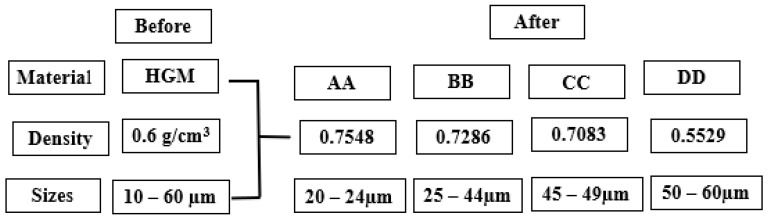
Schematic diagram for the particle size variation analysis by gas pycnometer.

**Figure 2 materials-16-01732-f002:**
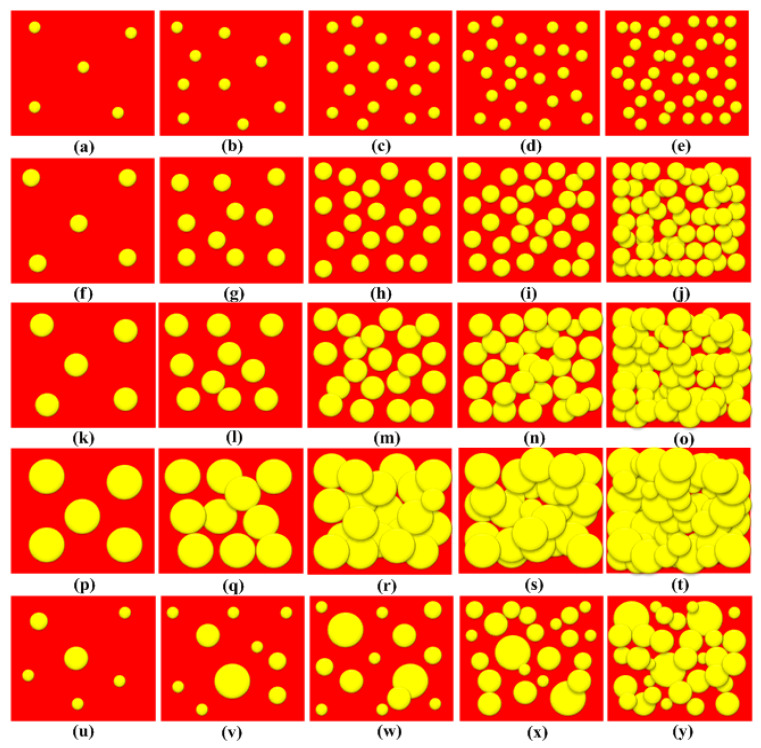
Schematic representation of microstructure for (**a**) AA5, (**b**) AA10, (**c**) AA15, (**d**) AA20, (**e**) AA25, (**f**) BB5, (**g**) BB10, (**h**) BB15, (**i**) BB20, (**j**) BB25, (**k**) CC5, (**l**) CC10, (**m**) CC15, (**n**) CC20, (**o**) CC25, (**p**) DD5, (**q**) DD10, (**r**) DD15, (**s**) DD20, (**t**) DD25, (**u**) OO5, (**v**) OO10, (**w**) OO15, (**x**) OO 20, and (**y**) OO25.

**Figure 3 materials-16-01732-f003:**
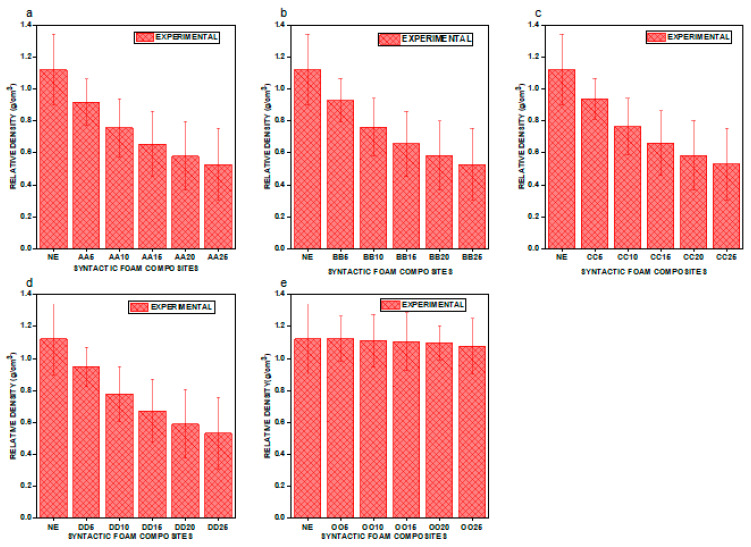
Graphical representation of density for (**a**) NE vs. AA5, AA10, AA15, AA20, and AA25; (**b**) NE vs. BB5, BB10, BB15, BB20, and BB25; (**c**) NE vs. CC5, CC10, CC15, CC20, and CC25; (**d**) NE vs. DD5, DD10, DD15, DD20, and D25; (**e**) NE vs. OO5, OO10, OO15, OO20, and OO25.

**Figure 4 materials-16-01732-f004:**
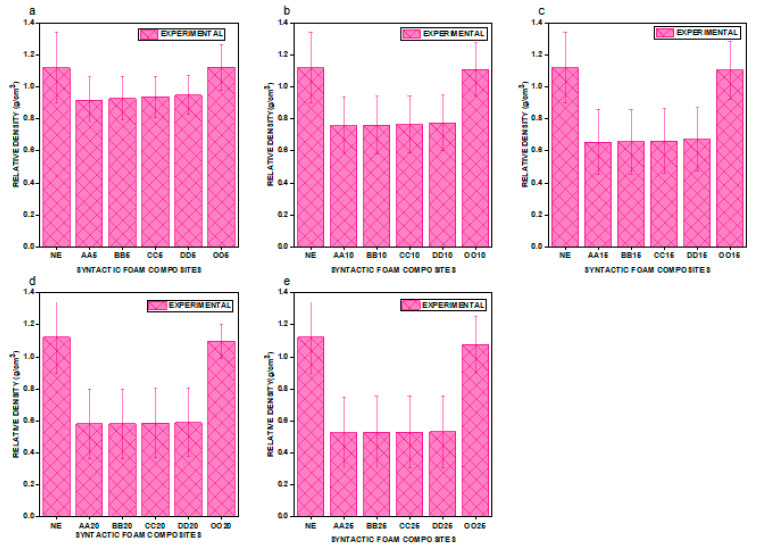
Graphical representation of density for (**a**) NE vs. OO5, AA5, BB5, CC5, and DD5; (**b**) NE vs. OO10, AA10, BB10, CC10, and DD10; (**c**) NE vs. OO15, AA15, BB15, CC15, and DD15; (**d**) NE vs. OO20, AA20, BB20, CC20, and DD20; (**e**) NE vs. OO25, AA25, BB25, CC25, and DD25.

**Figure 5 materials-16-01732-f005:**
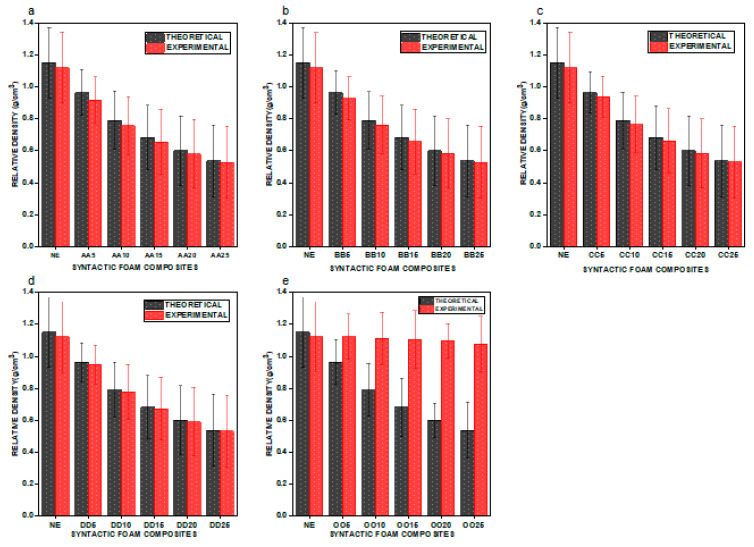
Graphical representation of experimental density versus theoretical density for (**a**) AA5, AA10, AA15, AA20, and AA25; (**b**) BB5, BB10, BB15, BB20, and BB25; (**c**) CC5, CC10, CC15, CC20, and CC25; (**d**) DD5, DD10, DD15, DD20, and DD25; (**e**) OO5, OO10, OO15, OO20, and OO25.

**Figure 6 materials-16-01732-f006:**
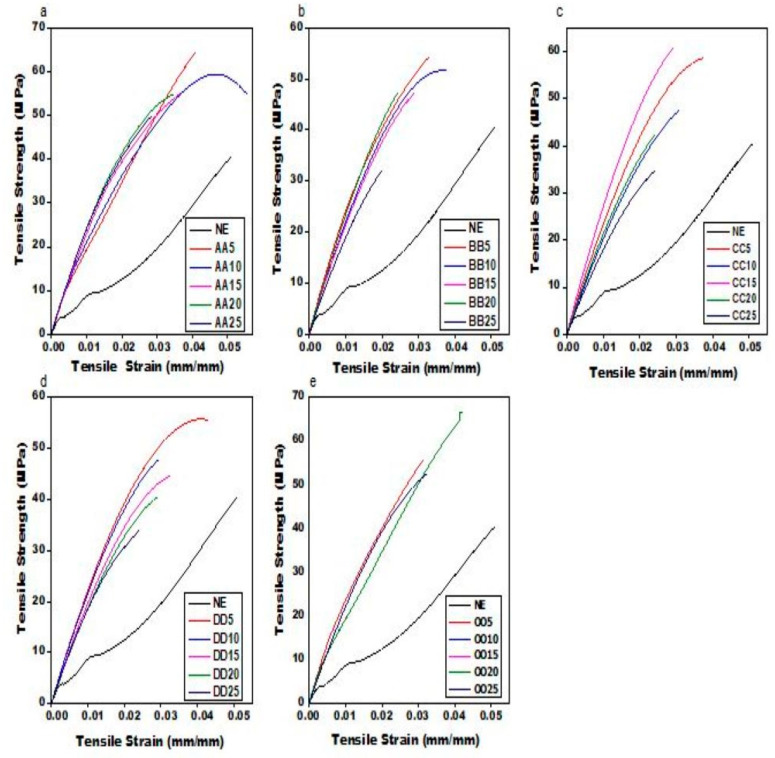
Graphical representation of tensile stress and strain for the comparison between (**a**) NE and AA5, AA10, AA15, AA20, and AA25; (**b**) NE and BB5, BB10, BB15, BB20, and BB25; (**c**) NE and CC5, CC10, CC15, CC20, and CC25; (**d**) NE and DD5, DD10, DD15, DD20, and DD25; and (**e**) NE and OO5, OO10, OO15, OO20, and OO25.

**Figure 7 materials-16-01732-f007:**
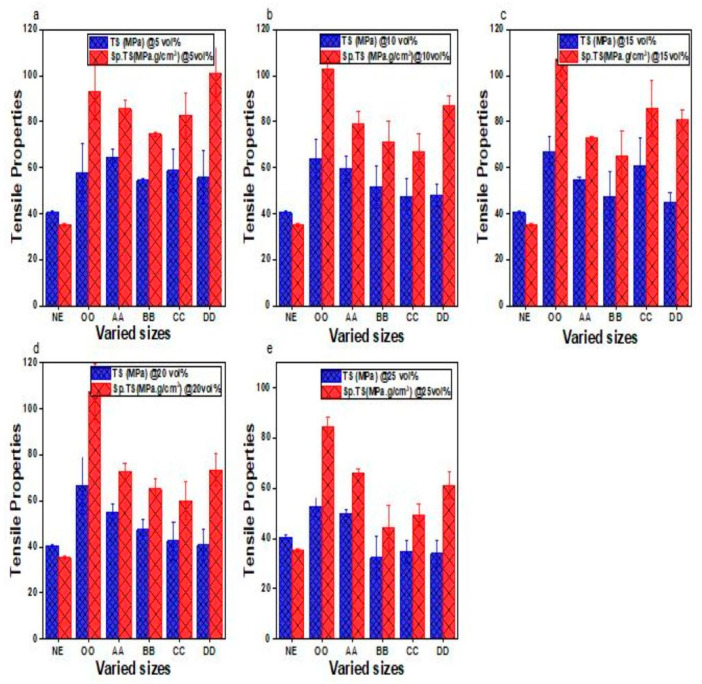
Graphical representation of tensile strength (TS) and specific tensile strength (SpTS) for NE, OO, AA, BB, CC, and DD: (**a**) 5 vol%; (**b**) 10 vol%; (**c**) 15 vol%; (**d**) 20 vol%; and (**e**) 25 vol%.

**Figure 8 materials-16-01732-f008:**
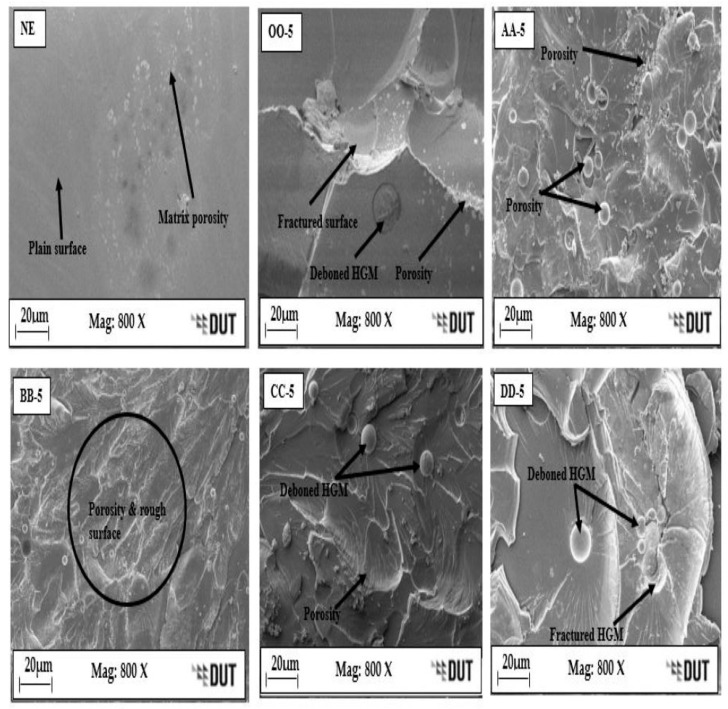
SEM images of the tensile fractured surfaces for the neat epoxy “NE”, OO–5, AA–5, BB–5, CC–5, and DD–5, showing matrix porosity, deboned HGM and rough surfaces.

**Table 1 materials-16-01732-t001:** Formulation for fabrication of e–SFCs.

Specimens	NE	AA5	AA10	AA15	AA20	AA25
		BB5	BB10	BB15	BB20	BB25
		CC5	CC10	CC15	CC20	CC25
		DD5	DD10	DD15	DD20	DD25
LR20	100	95	90	85	80	75
LR281						
HGM	0	5	10	15	20	25
Units	Volume percent (vol%)					
Code	AA: 20–24 µm; BB: 25–44 µm; CC: 45–49 µm and DD: 50–60 µm, OO: 0–60 µm, NE–neat epoxy					

**Table 2 materials-16-01732-t002:** Characteristics properties of syntactic foam composites with varied particle sizes and true density.

Size Variation	True Density $“\rho_{u}”$ (g/cm^3^)	Wall Thickness “t” (µm)	Aspect Ratio “a”	Tensile Strength @5 vol% (MPa)	Sp. Tensile Strength @5 vol% (MPa·g/cm^3^)
OO	0.6000	4.637	8.467	57.97	93.26
AA	0.7548	4.401	6.735	64.51	85.47
BB	0.7286	4.426	6.972	54.38	74.64
CC	0.7083	4.099	7.172	58.62	82.76
DD	0.5529	5.990	9.188	55.74	100.81

## Data Availability

Data for this research are not available due to ongoing research.

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
