# Peer review of "Processing of Low-Density HGM-Filled Epoxy–Syntactic Foam Composites with High Specific Properties for Marine Applications"

_materials, 2023, doi:10.3390/ma16041732_

Round 1

Reviewer 1 Report

Comments referenced by line numbers in the manuscript.

10-11. This sentence does not have sense.

16-17. 35% with regard to what? net resin? the lowest e-SFC?

41-43. Not clear

43-44. As expressed, it seems a general statement for all SFCs. Check if this is what you want to express, or you are referring to the particular material systems on the references [6,7].

44-46. Possibly referring to the  internal/external radius of the HGMs. But better if specified.

52-53. Not clear

61. Aspect ratio. It its explained later that is the d/t ratio, but at this point it is confusing, as typically, in reinforcing particles, aspect ratio refers to the higher to lower diameters.

Table 2. OO has not been defined. Is it the HGM before the sieve? If so, indicate and make it consistent the densities reported. In figure 1 it appears as 0.6 whereas in table 1 is 0.6216.

98-100. Without using colons is difficult to read.

102-103. Review. I do not see what “density or the diameter of the strength distribution” mean.

Figure 2. I do not find very useful this schematic representation, even can be confusing. For example, the number or circles are not representing properly the %V. Some images that are indicated as constant size particles you can find circles of different sizes.

124-125. Is M5, M10, M15, M20, and M25 using the OO HGMs?

133-135. Looks the same idea than in 129-132.

138. Wrong figure number

140-142. Insufficient analysis. The almost constant density in M5, M10, M15, M20, and M25 independently the volume fraction the is the strangest result in this work. Hence, it should be dedicated more effort to elucidate what it is the reason behind. I would say that formulating a hypothesis is not enough in this case.

154-155. Check reasoning, as the smaller the particle the higher is its surface/volume ratio. As the formulations are defined in volume concentrations, this would mean higher surface in contact with the epoxy for the smaller HGM sizes.

163-164. Packing efficiency, at volume concentrations lower that 25%, seem not to be a plausible reason. If the HGM particles are well dispersed in the epoxy, particles would have little contacts between them, as it can be observed in the included SEM images.

172. OO has not been defined.

174-175. Not clear

191-192. Hard to understand what this means.

220-224. The given explanation on the packing efficiency has not been supported by any micrographic analysis performed in the work. Neither, references by other authors have been included.

Author Response

The responses to the reviewers comment can be found in the attached document.

Reviewer 2 Report

Article" Processing of Low Density with high Specific Properties of

HGM Filled Epoxy -Syntactic Foam Composites for Marine Applications." is interesting but needs major revision before publication.

Comments:

1. The title should be corrected in its current form, it is incomprehensible.

2. Please complete the abstract with the numerical values of the measured density as well as tensile strength

3. In itroduction no references are made to publications from recent years on results obtained by other researchers on syntactic foam composites for marine applications.

4.Materials: Please specify the properties of the resin (viscosity, epoxy number) and hardener (type of chemical compound)

5.Fabrication of e-SFCs: On what basis did the authors choose post-cured at 80°C temperature?

6.Figure 2.:Requires more extensive commentary in the text

7.Tensile strength: please specify what type of samples were tested, beams or rowing boats, what were the dimensions when they were prepared.

8. Figure 7. The scale in the pictures is illegible

9. The conclusions are too general. They should be supplemented with specific values that have been improved and application aspects of the results obtained in terms of marine applications.

Author Response

The authors comments can be found in the attached document below.

Reviewer 3 Report

The manuscript under the title: “Processing of Low Density with high Specific Properties of HGM Filled Epoxy -Syntactic Foam Composites for Marine Applications” is in line with Materials journal. This topic is relevant and will be of interest to the readers of the journal. It based on original research. This research has scientific novelty and practical significance. The article has a typical organization for research articles.
Before the publication it requires significant improvements, especially:

  1. The "Introduction" section: it has been proven that the effect of various modifying additives and fillers on the physic-chemical and mechanical properties of epoxy polymer composites is determined by many factors: ……. I think the related references should be cited corresponding to each aspect, e.g. (but not limited to these), which will undoubtedly improve the "Introduction" section:

       - Polymers 2021, 13(15), 2421; https://doi.org/10.3390/polym13152421

       - Polymers 2022, 14(7), 1388; https://doi.org/10.3390/polym14071388

       - Inorg. Mater. Appl. Res. 7, 768–772 (2016). https://doi.org/10.1134/S2075113316050178

 2. Section 2.1. It is necessary to add the physicochemical characteristics of components - give a table with the main physicochemical and technological properties of epoxy resin, hardener and HGM.

3. It is necessary to add data on the change in the viscosity of the epoxy composition with the introduction of HGM

4.     4. Error bar should be added in Figure 3-6.

5.     5. Why are the maximum and minimum contents of the introduced filler (5% and 25%) so limited? It would be good to choose the minimum content HGM that provides the optimal reduction in the density of the composites.

6.     6. In my opinion, the filler content had to be increased until the introduction of HGM (OO) would not lead to a decrease in the strength properties of the composites. A higher filler content could significantly reduce the density of the composites.

7.     7. In my opinion, the increase in Fig.7. for samples should be the same, because it is incorrect to compare different samples at different magnifications.

8.     8. The mechanical properties which are significantly showing the efficacy of the modification, should be stressed more, in particular by showing some stress/strain graphic comparison, in order to determine also the elastic and plastic behavior modifications.

9.     9. The introduction of the original filler HGM (OO) provides the highest strength characteristics of composites, so why then fractionation?

Author Response

Authors responses to the reviewers comment can be found below.

Reviewer 3 comments and Authors responses.

The manuscript under the title: “Processing of Low Density with high Specific Properties of HGM Filled Epoxy -Syntactic Foam Composites for Marine Applications” is in line with Materials journal. This topic is relevant and will be of interest to the readers of the journal. It based on original research. This research has scientific novelty and practical significance. The article has a typical organization for research articles. Before the publication it requires significant improvements, especially: 1. The "Introduction" section: it has been proven that the effect of various modifying additives and fillers on the physic-chemical and mechanical properties of epoxy polymer composites is determined by many factors: ……. I think the related references should be cited corresponding to each aspect, e.g. (but not limited to these), which will undoubtedly improve the "Introduction" section: -

Polymers 2021, 13(15), 2421; https://doi.org/10.3390/polym13152421 -

Polymers 2022, 14(7), 1388; https://doi.org/10.3390/polym14071388 -

Inorg. Mater. Appl. Res. 7, 768–772 (2016). https://doi.org/10.1134/S2075113316050178

Response 1: Thank you for your comment and suggestions. The introduction section has been improved and the suggested articles referenced accordingly. Kindly see page 1, line 54 – 63; references 12, 17, and 18 in lines 345 – 346, and 357 – 361 respectively. 

  1. Section 2.1. It is necessary to add the physicochemical characteristics of components - give a table with the main physicochemical and technological properties of epoxy resin, hardener and HGM.

Response 2: Thank you for your comment and suggestion. The physiochemical characteristics of the component has been discussed in our previous work as referenced in page 2. Kindly see lines 80 – 82.

  1. It is necessary to add data on the change in the viscosity of the epoxy composition with the introduction of HGM

Response 3: Thank you for your comment and suggestion. The change in the viscosity of the epoxy composition with the instruction of HGM has been discussed in our previous work as referenced in page 3. Kindly see lines 95 – 96.

  1. Error bar should be added in Figure 3-6.

Response 4: Thank you for your suggestion. Error bar has been added to the suggested. Kindly see Figures 3-5, and 7, on pages 7, 9, 11, and  13 respectively.

  1. Why are the maximum and minimum contents of the introduced filler (5% and 25%) so limited? It would be good to choose the minimum content HGM that provides the optimal reduction in the density of the composites.

Response 5: Thank you for your comment and suggestion. The filler content was based on previous published work. Also, based on the application for consideration, increasing the filler content will increase the matrix dominant property and thereby create more voids which will affect the mechanical properties of the composite.

  1. In my opinion, the filler content had to be increased until the introduction of HGM (OO) would not lead to a decrease in the strength properties of the composites. A higher filler content could significantly reduce the density of the composites.

Response 6: Thank you for your comment and suggestion. Response 5 is applicable here also, nevertheless, your suggestion will be taking into consideration in the future.

  1. In my opinion, the increase in Fig.7. for samples should be the same, because it is incorrect to compare different samples at different magnifications.

Response 7: Thank you for your comment and suggestion. The same magnification has been used for proper comparison.

  1. The mechanical properties which are significantly showing the efficacy of the modification, should be stressed more, in particular by showing some stress/strain graphic comparison, in order to determine also the elastic and plastic behavior modifications.

Response 8: Thank you for your comment and suggestion. The stress-strain graphs for the mechanical property have been added  and discussed in page 12. Kindly see Figure 6, lines 207 - 226. 

  1. The introduction of the original filler HGM (OO) provides the highest strength characteristics of composites, so why then fractionation?

Response 9: Thank you for your comment. The fractionation of the sizes was done to experiment how reduction of density of SFC can improve the specific tensile strength of the material. Kindly see page 14, line 242-245.

Round 2

Reviewer 1 Report

In line 117, for me "average diameter of strength distribution" is confusing. I do no see what refers strength here. Perhaps "average diameter of the particles distribution" or "average diameter of the statistical distribution"? 

The same with "average density of the strenght distribution" in the same line.

Author Response

The authors sincerely appreciate your comments towards the improvement of this manuscript.

The comment/suggestion  has been corrected to "average diameter of the particles distribution" and " average density of the particle distribution" as suggested. Kindly see lines 116 - 117.

Reviewer 2 Report

Authors improved article with recommendation. Now is suitable for publication.

Author Response

The authors will like to appreciate the second reviewer for approving the manuscript for publication.

Reviewer 3 Report

The authors considered most of the comments or adequately responded to the remarks contained in the review; therefore, the work may be approved for publication.

Author Response

The authors sincerely appreciate the effort of the third reviewer for your meticulous comments and suggestions which has enhanced your approval for the manuscript publication.